

# Storm surge and extreme river discharge: a compound event analysis using ensemble impact modelling

Sonu Khanal[1,2,3], Nina Ridder[1], Hylke de Vries[1], Wilco Terink[2] and Bart van den Hurk[1]

[1]Royal Netherlands Meteorological Institute, 3730 AE De Bilt, The Netherlands
[2]FutureWater, Costerweg 1V, 6702 AA, Wageningen, The Netherlands
[3]Vrije Universiteit Amsterdam, De Boelelaan 1105, 1081 HV, Amsterdam, The Netherlands

*Correspondence to*: Sonu Khanal (sonu.khanal19@gmail.com / s.khanal@futurewater.nl)

**Abstract.** Many winter deep low-pressure systems passing over Western Europe have the potential to induce significant storm surge levels along the coast of the North Sea. The accompanying frontal systems lead to large rainfall amounts, which can result in river discharges exceeding critical thresholds. The risk of disruptive societal impact increases strongly if river runoff and storm-surge peak occur near-simultaneously. For the Rhine catchment and the Dutch coastal area, existing studies suggest that no such relation is present at time lags shorter than six days. Here we re-investigate the possibility of finding near-simultaneous storm surge and extreme river discharge using an extended data set derived from a storm surge model (WAQUA/DCSMv5) and two hydrological river-discharge models (SPHY and HBV96) forced with conditions from a high-resolution (0.11°/12 km) regional climate model (RACMO2) in ensemble mode (16x50 years). We find that the probability for finding a co-occurrence of extreme river discharge at Lobith and storm surge conditions at Hoek van Holland are up to four times higher (than random chance) for a broad range of time lags (-2 to 10 days, depending on exact threshold). This highlights that the hazard of a co-occurrence of high river discharge and coastal water levels cannot be neglected in a robust risk assessment.

## 1. Introduction

Floods are a major cause of casualties due to natural disasters, with over 6.8 million deaths globally during the 20th century (Doocy et al., 2013), and an annual average loss of 104 billion US$ (Blöschl et al., 2017). Rivers floods are the result of a complex chain of atmospheric and surface hydrological processes (Hall et al., 2014; Merz et al., 2014). Atmospheric circulation patterns distribute the necessary precipitation on a variety of spatial scales and intensities (Prudhomme and Genevier, 2011). Following the characteristics of the local hydrology the water then aggregates into streams and rivers, ultimately transporting river runoff to coastal estuaries. On its way to the sea, the main river is fed from several sub-catchments and receives input





from reservoirs that act on longer time scales than runoff routing processes. These long time scales introduce a dependence on antecedent conditions, for example ice and snowmelt from high-altitude glaciers and/or snow reserves located in the headwaters, soil moisture retention, and baseflow processes. In low-lying countries like the Netherlands, flood defense infrastructure protects the densely populated hinterland. Risk of river flooding therefore typically only occurs in the winter

season when deep Atlantic low-pressure systems (extratropical cyclones) precipitate large amounts of water over the European Alps and Central Europe.

Traditionally floods are defined by a set of correlated random variables like flood peak, flood volume, and duration (Brunner et al., 2016; Chebana and Ouarda, 2009). However, in the assessment of flood events this univariate approach to discharge is only suitable when dependencies between the contributing variables are known or absent (Serinaldi et al., 2015; Yue and

Rasmussen, 2002). Ignoring the dependencies may lead to severe over or under estimation of the (De Michele et al., 2005).

Coastal regions are threatened not only by river flooding. Storm surges and sea-level rise provide additional risk to flooding (Olbert et al., 2017; Wahl et al., 2017). Atlantic cyclones are prime candidates to cause both storm surges and large-scale precipitation over western and central Europe (Hawcroft et al., 2012; Ulbrich et al., 2009). Since storm surges find their origin in the very same Atlantic winter storms that also bring large amounts of precipitation, possible relations between the two

factors play a major role in the assessment of flood risks. If high coastal water levels occur simultaneously with extreme river discharge events, flood risk increases as the river is not able to discharge its water at the outlet, eventually causing river water levels to rise (van den Hurk et al., 2015). Further, projected sea-level rise will strongly amplify this risk (van den Hurk et al., 2014; Vries et al., 2014; Wahl et al., 2015). Previous studies investigated the probability of joint occurrence of storm surge and precipitation/river floods based on observed data along the British, Australian, and US coasts and Dutch coasts ( Geerse,

2013; Svensson and Jones, 2004; Wahl et al., 2017; Zheng et al., 2013). In literature, these sets of joint extreme events are categorized as compound events. Their societal importance and association to risk are well discussed by Hazeleger et al. (2015), Leonard et al. (2014) and Seneviratne et al. (2012).

Inclusion of tail dependence, i.e. the probability that a given variable exceeds a certain threshold given the exceedance probability of a threshold by another variable, is necessary for the investigation of joint extremes events (Joe, 1997). Poulin et

al. (2007), who used families of copulas to study the joint behavior of river flood peaks and flood volumes, demonstrated that resulting joint return periods are significantly sensitive to the choice of the copula model and inclusion of tail dependence in the analysis. Recent studies, such as Bevacqua et al. (2017) and Moftakhari et al. (2017) for the US coast and Italy respectively, further explored the importance of multivariate (copula) analysis of compound events related to coastal flooding including sea water level and river discharge. Also, these studies confirmed a substantial decrease in return periods if the compound

occurrence probability of related events was considered.

For the Rhine river catchment, previous studies have investigated those compound events based on observations, reanalysis products and model simulations. By using a coarse-resolution global climate model ensemble, Kew et al. (2013) showed that low pressure systems over the North Sea can lead to compound occurrence of extreme storm surges and precipitation affecting the Dutch coast. Proxies for storm surge and river runoff were used, namely north-westerly winds and multi-day precipitation





sums. In an attempt to use more realistic data to assess the statistical relationship between surge and river discharge, Klerk et al. (2015) subsequently used variables diagnosed from hydrological, hydraulic and storm surge models. In their coarse-resolution dataset covering the relatively short historical period from 1981-2010 they found a clear correlation between the two variables, but only when a substantial time lag of six days was taken into account. This is the time scale needed to transport

excess precipitation peak in the Rhine catchment to the river outlet. Accordingly, they conclude that there is no increased risk at simultaneous occurrence (zero lag). However, an in-depth investigation of the uncertainty introduced by the hydrological and storm surge model to the time dependence correlation was not performed.

Here, we build on the findings of the previous studies and extend them by including the application of (a) a fine resolution climate model, (b) a large sample of data (800 years) obtained from a climate model ensemble, and (c) two different regimes

of hydrological models. We believe that the use of a large sample of data obtained from a fine resolution climate model ensemble provides a better insight into the statistical connections at play between the two variables than possible in previous assessments which used samples from observations limited to the past 30 - 40 years only. Furthermore, by applying two hydrological models, we demonstrate the importance of proper physical model configurations to correctly assess the lag time correlation for compounding high coastal water level and high discharge events. Using a methodology developed by van den

Hurk et al. (2015), we force three impact models (one storm surge model and two hydrological models) with output from a large ensemble of a high-resolution regional climate model avoiding the necessity of proxies as applied in some of the previous assessments. Further, this approach provides us with a large dataset ensuring a solid statistical assessment of the problem. The combination of these improvements, i.e. the use of real impact variables rather than proxies, different regimes of discharge models and a longer timeseries obtained from a high-resolution climate model ensemble, improve our ability to draw more

consistent conclusions than possible in previous studies.

## 2.    Data and methods

### 2.1        Observations and climate model data

We use data from a 16-member ensemble obtained with the Global Climate Model (GCM) EC-Earth (Hazeleger et al., 2012), created using a perturbed atmospheric initial-state approach (start in 1850).  Each of the ensemble member is a plausible

representation of possible climate states and differs from each other in their initial atmospheric conditions and model internal variability. The internal variability of the climate system is highly nonlinear, which causes significant variability between model runs. Thus, each of the ensemble member represents how climate can vary due to chaotic internal variability (Deser et al., 2012).

This study focuses on the period from 1951 to 2000. The GCM ensemble is dynamically downscaled using the regional climate

model (RCM) RACMO2 at 12-km resolution (van Meijgaard et al., 2008). Daily precipitation output from RACMO2 is subsequently bias-corrected using gridded E-OBS V14 data at a 0.25° resolution (Haylock et al., 2008). The downscaled temperatures are further adjusted with a temperature lapse correction rate of -6.5°/Km. The dynamically downscaled and bias-





corrected data serves as input for two conceptually different hydrological models for the Rhine basin (a) SPHY (Terink et al., 2015) (b) HBV-96 (Bergström, 1976) and a storm surge model; WAQUA/DCSMv5 (Gerritsen et al., 2013). The performance of the hydrological models is tested against observed mean daily Rhine river discharge at Lobith, where the Rhine enters the Netherlands and daily mean total water levels (the sum of the tidal contribution and the non-tidal residual including the
meteorological effect referred to as TWL) at Hoek van Holland (HvH) for the period 1951-2000 provided by Rijkswaterstaat and are available online (www.waterbase.nl/).

### 2.2     Hydrological modeling of Rhine river discharge using SPHY and HBV-96

The application of two hydrological models allows the assessment of model-uncertainty in simulating discharge waves in relation to the natural variability due to e.g., dominant precipitation regions, and the sensitivity of different routing methods.
To this end, we use different hydrological models to simulate the daily discharge of the river Rhine at Lobith. From this point on the flow of the river is highly regulated with the main part of the river discharge being directed to the coastal outlet at HvH. Since snowmelt runoff plays a significant role in defining the hydrological regime and annual cycle along the Rhine (Stahl et al., 2016), we use models that explicitly account for this process.

The Spatial Processes in HYdrology model (SPHY) is a spatially distributed, raster-based "leaky-bucket" type model, which
operates on a gridpoint basis. It is a conceptual model that integrates parameterizations of the dominant hydrological processes: (i) rainfall–runoff; (ii) lake/reservoir outflow, (iii) cryospheric processes (snow, ice, glaciers), (iv) dynamic vegetation, (v) evapotranspiration, and (vi) root-zone moisture content. It contains sub-grid variability (e.g., cells can be glacier-free or partially to fully covered with glaciers) and is based on the widely-used degree-day melt modeling approach (Hock, 2003). SPHY requires daily precipitation, and daily maximum, minimum and average temperature as forcing input. SPHY was
calibrated against observed time-series of daily discharge (Obs), obtained from the Global Runoff Data Centre (http://grdc.bafg.de/), at seven locations along the Rhine for the period of 1989-1995. The calibration was performed sequentially; initially for five independent upstream (U/S) locations and subsequently for the two downstream (D/S) locations Andernach and Lobith. We used the mean square error (MSE) as the objective function and maximum likelihood estimation (MLE) criteria to calibrate the model parameters. Calibration and performance of the upstream sub basins are provided in the
supplementary material (Fig. S1, S2 and S3). The extreme flooding events of 1993/1994 and 1995, dominated by precipitation and snowmelt respectively (Engel, 1997), are included in the calibration period.

The second hydrological model is HBV-96 (hereafter referred to as HBV) originally developed by Swedish Meteorological and Hydrological Institute (SMHI, Bergström, 1976) and updated to version used here by Lindström et al. (1997). HBV is a "semi-distributed" conceptual model and, similar to SPHY, also includes parameterizations for processes such as snow
accumulation and melt, evapotranspiration, soil moisture and runoff. Unlike SPHY, HBV contains a detailed routing procedure to model flow between sub-basins and through lakes. In this study, we use the version of HBV calibrated by Deltares (Kamer et al., 2008). The final discharge series at Lobith for both the calibration/validation run and the 16-member ensemble were provided by Deltares.



### 2.3 Storm surge modeling of the North Sea using WAQUA/DCSMv5

The storm-surge model used in this study is WAQUA/DCSMv5 (hereafter referred to as WAQUA). WAQUA solves the two-dimensional shallow water equations on a $(1/8)°x(1/12)°$ grid to simulate water levels for the North Sea and its coastal areas. For this, the model takes into account the astronomical tide, the wind-induced movement of water and the barometric effect (Olbert and Hartnett, 2010) associated with the locally prevailing sea level pressure. Therefore, when a low-pressure system travels over the North Sea, WAQUA responds to the main meteorological forcings that affect shallow seas during such an event. Tidal effects, modified in amplitude and timing by geometry of the coast and the underlying bathymetry, are calculated separately to this meteorological forcing. To obtain the TWL, WAQUA is first run using only the harmonic components at the domain boundaries, while the meteorological forcing is neglected. In a second step, WAQUA uses the calculated tidal level from the previous step and applies the meteorological forcing to calculate TWL. The non-tidal residual (hereafter referred to as surge) is then derived by subtracting the tidal level from the TWL.

A detailed description of how surge and tides are computed in WAQUA can be found in Gerritsen et al. (2013). The model sensitivity and capability to represent relevant air-sea momentum transfer processes and annual extreme water levels is described in Ridder et al., (2018). For the study presented here, the daily mean TWL (tide plus surge) at HvH is used.

### 3. Performance of the surge and discharge models

It is important to assess the basic quality of the surge and discharge models. For a validation of the storm surge model WAQUA the reader is referred to previous studies that present this assessment in detail (van Meijgaard et al., 2008; Ridder et al., 2018). Since the aim of this study is the exploration of relations between river discharge at Lobith and coastal water level at HvH, the amplitude, timing and duration of (extreme) discharge events are particularly critical.

### 3.1 Hydrological models (SPHY and HBV)

To assess the performance of the two hydrological models, both HBV and SPHY were forced with bias-corrected E-OBS daily precipitation and temperature data for the period 1951-2000. The output of both models thus produced (hereafter referred to as EOBS runs) is compared to the observed discharge at Lobith. The assessment focuses on the ability of the models to reproduce discharge amounts, duration of the flood wave (Sect. 3.2.2) and timing of flood peaks (Sect. 3.2.3). For this we use a simplified threshold approach to identify the flood waves in the discharge time series. In this study, a flood wave event is defined as a series of consecutive days with daily discharge exceeding the 95[th] percentile of the annual distribution. The length of each flood wave (in days) is calculated as the time difference between the onset (flow exceeds threshold) and offset (flow falls below threshold). The 95[th] quantiles are computed independently for the observations and the models based on the respective full-time series to account for model biases.





### 3.1.1 Basic metrics and distribution

Fig. 1 shows the daily Rhine river discharge for the entire period 1951-2000 as modelled by HBV and SPHY compared to observations. HBV tends to slightly overestimate observed high peaks (flow greater than 95[th] percentile) by up to 800 m3/s on average (> 3000 m3/s in some cases). SPHY, in contrast, significantly underestimates discharges at high thresholds. Modeled

discharge in SPHY does not exceed 8000 m3/s, making it unsuitable as a forecast model for Rhine river discharge extremes without additional statistical post-processing (such as quantile correction). Nevertheless, both models show the ability to represent observed discharge values fairly well with an overall bias and Nash Sutcliffe Efficiency (NSE, Nash and Sutcliffe, (1970)) of 1.3% and 0.72 for SPHY, and -10.3% and 0.87 for HBV (Table 1). HBV thus outperforms SPHY at all metrics except bias. However, SPHY is slightly better in reproducing the observed seasonal cycle of discharge particularly at lower

percentiles with the largest values occurring in winter (suppl. Fig. S4).

The normal quantile-quantile plot using observed and modelled discharge (Fig. 2) shows that both HBV and SPHY underestimate the observed flow at low discharge values. At intermediate values (ranging from -1.5 standard deviation to +1.5 standard deviation of mean discharge) SPHY reproduces observations fairly well, while HBV clearly outperforms SPHY for values above 2 standard deviations. At the highest discharge extremes, observations lie in-between the discharges of HBV and

SPHY with HBV following observations more closely.

To investigate the possible cause for the systematic underestimation of the high-extremes at Lobith in SPHY, we investigated the scatterplots for the individual sub-basins as shown in the supplementary material (suppl. Fig. S2). The discharge at upstream locations (Untersiggenthal, Reckingen, Rockenau, Cochem and Frankfurt) matches observations more closely than at the two downstream locations, Andernach (see suppl. Fig S2 (f)) and Lobith (see Fig. 1). This suggests that the model's shortcomings

at Lobith originate in the aggregation of sub-catchments. The simplified routing scheme may be responsible for this bias. SPHY uses a simple flow recession coefficient (kx) to simulate the delay between generation of specific runoff within the catchment and reaching the outlet. The recession coefficient is used as a weighting parameter to calculate the routed flow at each pixel, which in SPHY is calculated as weighted average inflow of the current and previous day (Terink et al., 2015). The averaging used in SPHY causes the attenuation and delay of the floodwaves in the model. In contrast, HBV uses a much more

complex routing scheme (Hegnauer et al., 2014). The differences in routing scheme have a significant impact on the floodwave length and peak.

### 3.1.2 Flood wave duration distribution

Using the definition of a flood wave event described in Sect. 3.1, i.e. flows higher than the 95th percentile of the respective dataset, we find 92 (SPHY), 113 (HBV) and 116 (observations) flood waves in the 50 years between 1951 and 2000, just under

two per year on average. Most of these events have occurred in the winter half year (October to March). Their duration ranges from one to up to 35 (in HBV), 40 (in Obs), and 50 (in SPHY) days (Fig. 3-left). As a result, HBV reproduces the observed mean flood wave length of approximately eight days fairly well while SPHY overestimates the mean duration. Note that the



calculation of the mean flood wave length is based on the number of flood waves in the respective dataset. Therefore, in SPHY fewer occasions of overestimated flood durations affect the mean of the flood wave length more strongly than in HBV where a higher total number of flood waves can mask the effect of prolonged flood waves. This can be seen in the empirical cumulative distributions (ECDFs) of the flood duration for the hydrological models and observation (Fig. 3-right). The ECDFs

of the two models show that half of the events (ranging from a probability of 0.25 to 0.75) last between 4-9 days in HBV and 2-9 days in SPHY (Fig. 3-right). Hence, there is no consistent overestimation of flood wave duration in SPHY but the overestimation of the mean is highly biased by a few long flood waves.

Adding the ECDFs of the different RACMO2 ensemble members (light blue and red lines) highlights the range of uncertainties in flood wave length. A comparison between ensemble members and observations is of course not possible. However, adding

them alongside the ECDFs of the two EOBS runs (blue and red solid lines) gives an indication of the influence of random internal variations (see Sect. 2.1) and highlights that the results of both models fall within the physically plausible flood wave durations.

### 3.1.3    Timing of onset and peak

In compound events, two or more variables reach high percentiles of their distribution simultaneously, or in rapid succession.

Therefore, the ability of hydrological models to reproduce the timing of events in impact-parameter space (TWLs and discharge) is crucial. However, neither SPHY nor HBV are able to reproduce all observed flood peaks. Out of the 200 highest maximum discharge dates in the observations, 141 are realized in HBV, but only 96 in SPHY. A better match is reached if we allow for some flexibility in the timing, accounting for random errors in the meteorological forcing or runoff generation process. The modelled arrival time of the peak of a flood wave is sensitive to many parameters, including intensity, duration

and frequency of precipitation, soil moisture, the existence of basal flow (e.g. due to snow/ice melt upstream), river bathymetry and topography. Modeling and representation errors in each of these processes can explain part of the large difference in observed and modeled peak-flow days.

An accurate match of the modelled timing of maxima to observations is difficult, especially in SPHY. The flood waves in SPHY usually have multiple maxima which makes it difficult to correlate with the exact peak as found in observations. This

is further exacerbated by the consistent mismatch between observed and modeled peak timings. To overcome this, we use the onset of floodwave as a proxy of the peak of the floodwave. The onset day is defined as the day when the discharge value first exceeds the threshold ($95^{th}$ percentile) of the respective distribution (Sect. 3.2). The impact model onset dates are then compared with the observed onset dates allowing for a difference of ±5 days to match. Since, taking longer than ±5 days might result in matching the flood peaks entirely from different events, we chose it to use ±5 days here. From 116 floodwaves in

total, floodwaves completely missed by the impacts models, i.e. those that are either not realized in the modeled timeseries (8 in HBV and 32 in SPHY) or those that lie outside (12 in HBV and 10 in SPHY) of the ± 5 days range, are discarded. Fig. 4 shows the distribution of arrival-time differences in onset of flood. Both models mostly have a negative time lag (implying that the most peak arrive earlier than in the observations) but in HBV the negative time lag (about one day) is smaller than the

 

2-3 day lag in SPHY. This suggest both the models have errors in estimating the timings of flood waves and SPHY being the worst of two. Half of the data points lie between -1 and 1 for HBV and -4 and 1 for SPHY. The presence of positive values in the distribution indicate the models waves sometimes occur too late. The broad shape of the distribution of both HBV and SPHY reflects the complex interaction of the climatic and hydrologic processes.

In summary, SPHY strongly underestimates the peaks, while HBV is performing rather well except for an overall negative bias of the discharge. However, the mean seasonal cycle is better represented by the SPHY than HBV. Both models have difficulties in reproducing flood timing, and SPHY and HBV generate flood waves that occur systematically too early. However, the distribution of timing errors is quite broad which makes correcting for this error difficult.

## 4. Dependencies between storm surge and river discharge

In this section, we assess the dependence between Rhine river discharge at Lobith (modeled by SPHY and HBV) and TWL at HvH (modeled by WAQUA). Since most storm driven high water levels and discharge events occur in winter, we focus our analysis on the winter half year only (October - March). Here we repeat the earlier analyses of Geerse, (2013) and Klerk et al. (2015), but account for a range of both positive (discharge event succeeded by TWL event) and negative (discharge event preceded by the TWL event) time lags in our assessment. A positive lag can account for the time required for the weather

system to move inland including hydrological responses of the catchment. A negative lag may result from an unusual track of the passing weather system, or from natural variability where the events are shaped by a sequence of storms. The use of a range of time lags allows also to account for the inherent model uncertainty in reproducing the correct wave travel speed (Fig. 4). To analyse the influence of the choice of time lag on the dependence structure of high discharges and high TWL, we base our assessment on extreme values in TWL. For this we define a high/extreme water event as a day where the daily mean TWL

at HvH exceeds the 90th percentile of its distribution.

Fig. 5 shows this composite for the RACMO based ensemble of WAQUA-HBV and WAQUA-SPHY. In absence of any correlation between TWL and Rhine discharge, the 90%-confidence bands of the percentiles of the lagged time series around a high-water event (shaded area) would overlap with the unconditional discharge percentiles. This is not the case. Depending on the hydrological model, and more importantly, depending on the discharge percentile considered, the composite shows

elevated discharge levels for a range of lags. For the 90[th] percentile the discharge levels are significantly elevated at lags ranging from -2 days to >10 days. This indicates that there is dependence between the two variables starting two days before a high water (90th percentile) event and lasting up to ten days after the event. The elevated conditional discharge two days before a high TWL event possibly due to the twin or series of storm surge where the water level is already elevated by the first storm event. These ranges of lag are consistent with the time (i) the low-pressure system causing the conditions in both

variables requires to move over the catchment area, affecting the fetch for surge and precipitation starting locally and moving further upstream, and (ii) the transformation of rainfall to runoff and the propagation of the runoff waves to the downstream location of Lobith, and (iii) natural variability. HBV shows significantly elevated discharges at the 99% quantile for a lag of





four to eight days. In contrast, the discharges at this quantile in SPHY show no significant difference from climatology. This is understandable given the strong cutoff of the discharge distribution in SPHY (Sect. 3). The higher quantiles in SPHY, especially for peak discharges, are broadened and show multiple maxima. Therefore, the shortcoming of SPHY to underestimate observed high quantiles is also reflected in the lagged discharge composites. However, quantiles 90th and 95th

show a clear deviation from the climatology.

### 4.1        Dependence in the tail of the distributions

In this section, we examine how the upper tails of these two distributions are related. As mentioned in the introduction, tail dependence has been shown to be of particular importance for the influence of compound events on flood risk. Since tail dependence describes the degree of dependence in the tails of a multivariate distribution, we investigate the tail of the discharge

distribution conditioned on the distribution of total water level and compare it with the unconditional tail of the discharge distribution. The result of this analysis determines whether or not there is any underlying correlation in the tails of the discharge and TWL distributions. The availability of 800 years of data provides us with sufficient confidence in analysing the tails of the tail of distributions.

Having determined the correlation for a range of time lags in discharge and TWL (Fig. 5), we now examine the dependence in

the tail of the distribution for a lag of three days, i.e. with a water level event at HvH occurring three days before the discharge peak at Lobith. This is a shorter time lag than applied by Klerk et al. (2015), who presented the dependence in the tail for a lag of six days. Despite this shorter time lag we find a similar relation between the two variables as Klerk et al. (2015) (Fig. 6a and 6c). For this, we test the 50th and 90th percentile of the discharge distribution conditional on TWL and compare it to the respective unconditional discharge. We use seven different percentiles of the full water level distribution to determine the

conditional discharge (50, 60, 70, 80, 90, 95 and 99). The conditional discharge values lie above the unconditional discharge. This suggests that the two variables show some correlation in the tail of their distribution even at a lag shorter than the six days.

Since the dependency in tail of these two-distribution is already realised at a lag of three days, we also assess if the correlation still holds for other choices of lag duration. For this, we investigate the discharge distribution conditioned on specific (50th and

90th) quantiles of TWL. We find that, due to the longevity of floodwaves and the natural variability within the system, this tail dependency can be shown to hold for a range of lags. To illustrate this, we chose the 50th and 90th quantile of TWL and again determined the part of the discharge distribution between between 50th and 90th quantiles, i.e. the high tail of the conditional discharge distribution. Fig. 6b and 6d show the discharge conditioned on the 50th and 90th percentile value of TWL as function of time lags varying from -15 to 15 days. Two important aspects can be seen in this figure: (1) the width of the band increases

with time lag, and (2) the timescales at which the band approaches the climatology, i.e. none of the ensemble members shows a correlation between discharge and TWL. The band width (black and blue area) represents the conditional discharge distribution (part of distribution between the 50th and 90th quantiles) for the respective TWL quantile. The increasing band width suggests a large variation in the distribution (between 50th and 90th quantiles) and vice versa. The band width is



decreasing with increasingly negative lag days and approaches its climatological value quickly. This implies that discharge and TWL before a high coastal water level event are uncorrelated and their connection is not different than during normal climatological conditions. For positive lags the band width increases with increasing number of lag days and attains a maximum deviation from climatology around 4-8 days. This variability in bandwidth could be explained by the physical connection

between TWL and river discharge for positive lags and indicates that both variables have the same origin, i.e. winter storms building high surge levels at the coast and dropping large amounts of precipitation to the catchment resulting in high river discharge levels after few days. The 90% confidence interval within the ensemble (i.e. $5^{th}$ and $95^{th}$ quantiles) in bandwidth and are calculated from the ensemble. The ensemble confidence is larger in estimating the bandwidth for $90^{th}$ quantile (grey band) than the $50^{th}$ quantile (blue band). Like bandwidth, ensemble spread (bars) increases for 4-8 days positive lag, but are still

higher than climatology. Conversely, at negative lags the uncertainties overlap with climatological values. The uncertainty bands are higher than climatology only after positive lag of two days for both HBV and SPHY.

Other than for negative lags, the bandwidth for large positive values does not reach the climatological values. This is mainly imposed due to the slow hydrological response of the catchment and large duration of the river floods. This means that local precipitation at the discharge location occurs within hours, while the precipitation over the upper regions of the catchment

takes several days to be transformed into river discharge and travel downstream. Furthermore, in the time upstream precipitation takes to reach the discharge location a second low pressure system closely following the system causing the initial increase in river flow, can elevate the discharge through precipitating water in the same downstream region.

### 4.2 Joint distribution

In order to discard the possibility that the joint occurrence of high discharge and water level (red dots in Fig. 6a and 6c) are

based on random chance, we follow the method applied in Kew et al. (2013) and van den Hurk et al. (2015). For this, the probability density function (PDF) of the full, physically related ensemble is compared to a "randomized version" where the statistical relation between the two variables is removed by combining random pairs of variables (Fig. 7). Results are shown for a three-day lag, but similar results are found for all lags where tail-dependence was shown (see Sect. 4.1).

The shuffled dataset is derived from the original data in the following way. First, the 16 discharge ensemble members were

shuffled (changing the order of ensemble randomly). Then this new data is paired with the TWL data from all other ensemble members. In this way fifteen shuffled sets of 800 years of paired data were created. The reference data set, in which the physical between the two variables is retained, links TWL to the discharge three days later. Maximum discharge from the reference and shuffled data sets are shown in Fig. 7. Each ensemble member is shown using a different colour. Both hydrological models show similar joint distribution patterns for the reference dataset. However, HBV reaches further into the high discharge area

of the distribution than SPHY. This is to be expected due to the shortcomings of SPHY to reproduce high discharge events (Sect. 3.2.1).

The area of interest for the assessment of the connection between coastal water level and discharge is the area between quantile contours of the reference and randomized joint probability distributions. Consistently probabilities of joint occurrence of high



TWL and discharge levels are found in the reference data set exceed random chance in the diagonal top direction, which indicates an increasing dependence with increasing return period. Conversely, combinations of high TWL/low discharge and vice versa are less likely than random chance (diagonal perpendicular to the previous diagonal). This elongated shape of the joint distribution of the reference data along the diagonal indicates a positive correlation between the TWL and discharge

volumes three days later. The same diagram for other lag days shows similar features suggesting that this correlation is not limited to one fixed choice of lag but exists over a range of time scales. The main differences between the different distributions depending on lag day is that the correlation is stronger for lag days around six days with the regions only covered by the reference distribution being bigger than for shorter time lags (suppl. Fig. S5).

### 4.3    Compound probabilities

Up to now we have shown several aspects of the relation between lagged extremes in TWL and Rhine river discharge at Lobith. In this final section we compare the probability of Rhine discharge exceeding a certain level, given that TWL at HvH is also high. The metric of interest is the conditional probability scaled with its unconditional probability. For example, if TWL and discharge were completely uncorrelated, we expect a random probability x*y that TWL > $x^{th}$ percentile and discharge > $y^{th}$ percentile of their respective distributions. By scaling the probabilities by the unconditional probability, the inflation factor

due to the tail-dependence is quantified. Fig. 8 shows these scaled probabilities for TWL > 97.5% percentile of the winter distribution, and a range of discharge exceedance probabilities. Results are shown for various time lags. A value of one implies probabilities not exceeding the random chance probability. These values near unity are found at all lags for the lowest discharge percentiles. However, for increasingly rare conditions (higher discharge percentiles), the normalized probabilities strongly increase. For HBV up to 2-5 times higher values are found for the higher discharge percentiles, for a broad range of time lags.

This implies that it is 2-5 times more likely to get a high discharge once TWL is high. SPHY shows qualitatively the same result. Although its peaks reach lower values, the levels remain elevated for longer positive lags than in HBV. This is clearly related the generally longer duration of the floodwaves in SPHY.

### 5.    Discussion

Using the dynamically downscaled output of the EC-Earth ensemble, we obtain coastal water levels and associated river

discharge from one state-of-the-art storm surge model (WAQUA/DCSMv5) and two hydrological river-discharge models (SPHY and HBV). The use of a physically related 16-member ensemble provides us with a larger sample size (800 years), allowing a solid statistical assessment of physical relationship that cannot be carried out with observations alone, as applied in previous studies. The increased sample size is particularly important for the assessment of the high tails of the tail of two distributions and their joint distribution.

This study is limited by the performance of the applied models. The results presented here carry the bias of the meteorological and hydrological models. For instance, both hydrological models show discrepancies to observed high discharges. While HBV



overestimates the discharge at high values, SPHY tends to underestimate it. These large biases mainly emerge in the peak discharge events, characterized as multi-day events occurring roughly twice every year. The mean discharge ($50^{th}$ quantile) and higher quantiles ($90^{th}$, $95^{th}$ and $99^{th}$) are represented reasonably well in both models (suppl. Fig. S4). The same is valid for the storm surge model WAQUA, which has been shown to reproduce observed coastal water levels and their extremes fairly well (Ridder et al., 2018). Since the results presented here are based on quantile thresholds relative to the respective dataset, the biases in the model results do impact the findings concerning the statistical relation between the variables water level and river discharge. The results of two hydrological models with opposite bias allows evaluation of the impact of these bias on the correlation characteristics, and gives an indication of the contribution of model bias to uncertainty of this joint correlation.

The presented results are based on daily values, both in the analysis of the relevant variables as well as in the forcing of the hydrological models. Consequently, the data does not contain any diurnal variations in river discharge and surge/TWL. Averaging to daily values leads to smoothening of the response and to additional uncertainty in the timing of the peak onset and duration. Moreover, the effect of two or more consecutive precipitation events within a couple of days (analogous to two or more small depression passing over in quick succession) could be considered as one. This would result in underestimation of extreme events frequency, particularly in hydrological sense. Since with the current threshold approach to identify the flood wave, differentiation between the flood wave generated separately from multiple precipitation event in quick succession is not possible. More importantly, two storms in quick succession ("twin storms") where the second storm coincides with delayed high runoff could not be resolved completely and considered as one in the present setup. Although, the elevated conditional discharge 2 days before a high TWL event, to some extent, is explained by "twin storms" still it's not completely understood in this study. While it is unlikely that this has a significant impact on the probabilities of occurrence of compounding events it should be highlighted as it may lead to a (small) underestimation of the number of these events.

The impact models, especially SPHY, are unable to resolve a distinct peak at higher quantiles. To avoid a large mismatch of timing of floods peaks, we chose to define the onset of floodwave as a proxy of the peak of the floodwave. We found both hydrological models are unable to simulate the correct timing of the flood peaks, but flood durations are well represented in both models. The use of a simple recession based routing method is presumably responsible for errors in the correct timing of the floodwave in SPHY. This is likely to be improved when a cascade of hydrological and hydrodynamic (hydraulic) models is applied instead of the single hydrological model configuration as applied here.

The uncertainty due to downscaling from RACMO may introduce spurious biases in the results presented here. RACMO is known to release precipitation too close to the coastline (van Meijgaard et al., 2012) which makes it difficult to estimate the basin scale hydrological responses of the synoptic scale circulation pattern. This may lead to an overestimation of the local flood magnitude near coastal areas. Additionally, the most extreme wind speeds can only be generated at scales that are finer than those resolved by RACMO. Consequently, the modelled TWL may display a bias as well. The statistical bias correction methods do not always accurately preserve the properties of extremes and associated signals (Christensen et al., 2008; Ehret et al., 2012; Sippel et al., 2016). The possible alteration in extreme precipitation signals, due to bias correction of downscaled EC-Earth ensemble, may affect the results presented here.



Nonetheless, we show the joint/compound events, i.e. high discharges and high TWLs, are physically related to each other. This is an important finding to improve assessments of coastal flood risks.

## 6.    Conclusion and recommendation

The temporal dependence structure of compounding extreme coastal water level and river discharge events has been analysed. Taking into account model uncertainty, natural variability and duration of flood peaks, we find that correlation between the discharge at Lobith and coastal water levels at Hoek van Holland occur at a range of time lags. Other than previous studies, which are based on historical observations, we find model uncertainties make it necessary to consider a range of lag days rather than a fixed time lag of six days. Thus, considering these uncertainties, the impact of co-occurring high river discharges and coastal surge levels cannot be neglected for large catchment areas as the Rhine basin, as was suggested in earlier studies. Our results show that even shorter time lags show significantly increased probabilities of joint occurrence. Neglecting these short time lags can lead to significant underestimations in return periods and thus flood risk. Finally, this study illustrates the importance of good physical models in compounding event analysis framework to allow for solid and reliable assessments of the events.

It is possible that the temporal dependence structure will be significantly altered when different subcomponents of discharge such as rain, snow melt and baseflow are considered separately. While we did not assess this in the present study, we believe it to be a valuable assessment and plan to follow up this line of investigation in a future study.

*Code and Data Availability*. The code of the SPHY model is publicly available at https://github.com/FutureWater/SPHY. The WAQUA model codes are available upon request. The datasets that are produced in this study are available upon request.

*Author Contributions*. The study is designed by Bart van den Hurk, Hylke de Vries, Nina Ridder, Sonu Khanal and Wilco Terink. The SPHY model was run by Sonu Khanal and Wilco Terink. The analyses of the results were performed by Sonu Khanal,  Hylke de Vries and supported by Nina Ridder. Sonu Khanal, Nina Ridder and Hylke de Vries prepared the manuscript text. The figures were prepared by Sonu Khanal. Finally, the proof-reading was performed by all (co-)authors.

*Competing Interests*. The authors declare that they have no conflict of interest.

*Acknowledgements*. This project has received funding from the European Union's Horizon 2020 research and innovation programme under the Marie Skłodowska-Curie grant agreement No 676027. This study was also partly funded by the Netherlands Organisation for Scientific Research (NWO) as part of the project "Impacted by Coincident Weather Extremes" (ICOWEX; grand number 869.15.017). The authors would like to thank the IMPREX project, in particular Femke Davids and



Erik van Meijgaard, for providing the data from two of the impact models. We also thank Ferdinand Diermanse for his valuable comments and expert advice during the data analysis for this study.

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





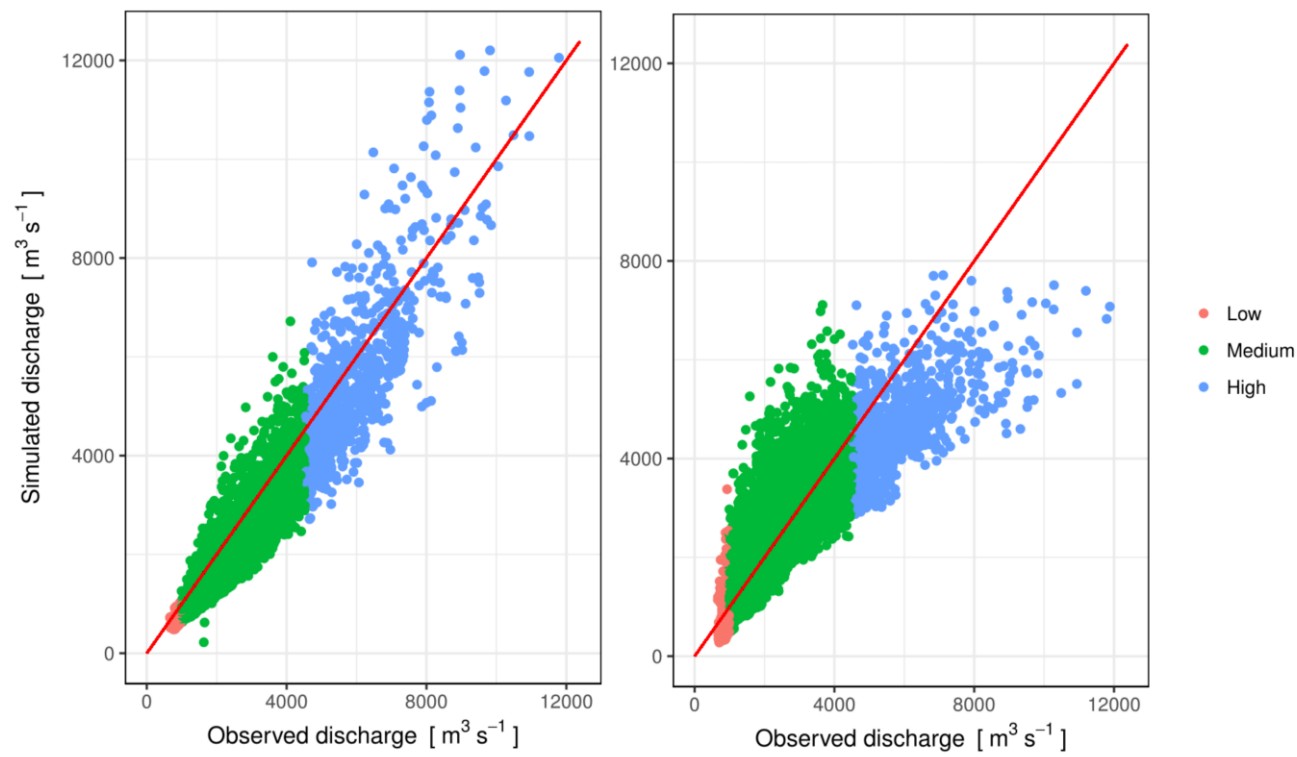

**Figure 1:** Observed versus modeled daily discharge at Lobith for the period between 1951 to 2000. (left) HBV, (right) SPHY. Colors indicate three ranges based on observed percentiles: "Low" (<5%, red), "Medium" (5-95%, green) and "High" (>95%). The solid red line
10   represents the 1:1 slope.





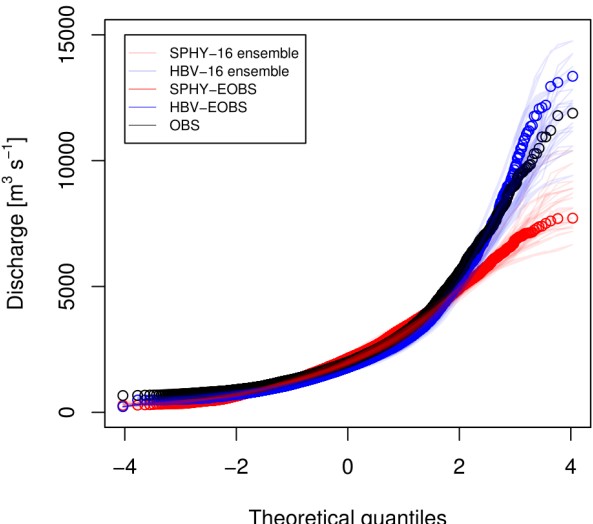

**Figure 2:** Normal quantile plot for HBV (blue), SPHY (red) and Observation (black). On the horizontal axis, the distributions are centered and scaled (divided by the standard deviation). The light blue & red lines represent 16 ensemble members for HBV & SPHY.

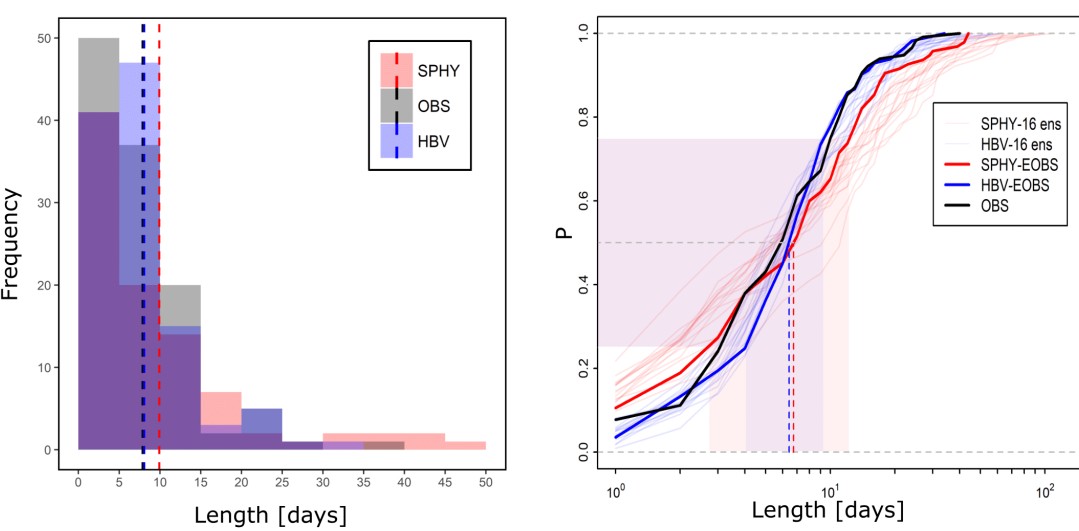

**Figure 3:** (left) frequency plot for the flood wave length above 95th quantile threshold of discharge. The dash vertical lines show the mean of the flood waves for SPHY (red), HBV (black) and HBV (blue). The Empirical cumulative distribution function (ECDF) plot (right) for the floodwave length above 95th quantile threshold of individual time series discharge. The light red and the blue shaded area represents the 0.25 and 0.75 for SPHY and HBV respectively.





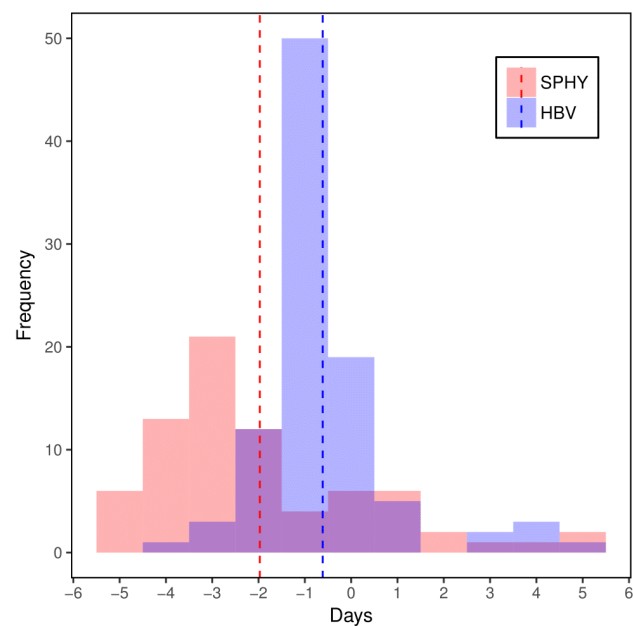

**Figure 4:** The frequency plot for the onset of the floodwave models compared with the observation. The X axis represent the lag in the onset
10  of the wave as compared to the observed onset. The negative & positive value represents the onset of the modeled waves are earlier & later
than the onset of observed floodwaves. The dash line (blue & red) represent the weighted mean of the distribution (HBV & SPHY).





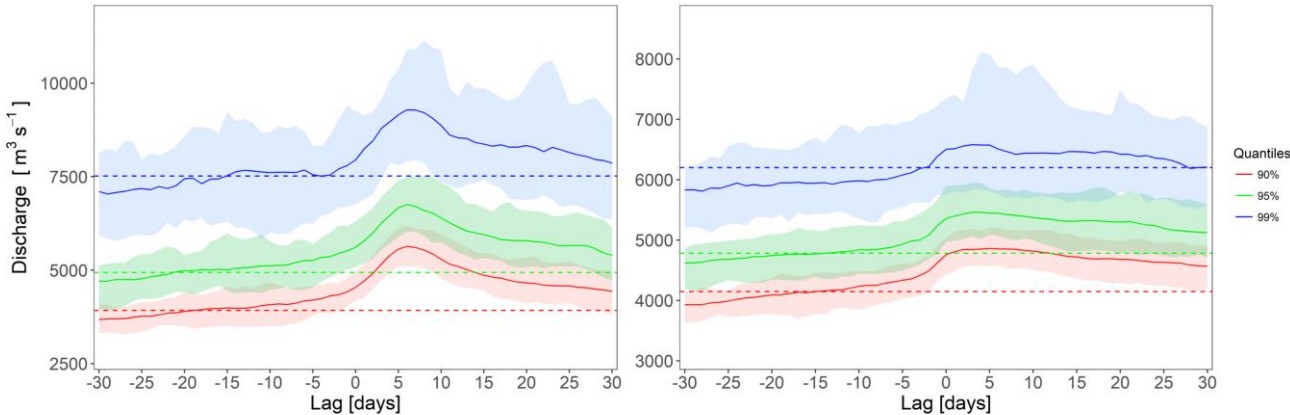

**Figure 5:** Mean temporal evolution of the 90th (red), 95th (green) and 99th (blue) quantile of discharge at Lobith for total water level events exceeding the 90th percentile at HvH in WAQUA as modeled by HBV (left) and SPHY (right). The lag in discharge at Lobith is relative to the peak in total sea water level at HvH. Negative & positive lag days indicate that the discharge peak occurs before & after the day of the high sea water event. The dashed lines are the unconditional discharge quantiles, i.e. discharge quantiles independent of water level; solid lines are the ensemble mean of the conditional quantiles. The shaded area represents 16 different lines for each ensemble and we only took the 5th and 95th percentile of those 16 lines to show spread of 16 ensemble member.







**Figure 6:** Left: Scatter plot of coastal water levels and discharge for a lag of three days ((a) HBV and (c) SPHY) and 16 ensemble members. Events exceeding the 99th quantile of either of the variables are marked in blue. Events exceeding the 99th quantile of both variables are marked in red. The triangles (blue/black) represent the ensemble mean of the conditional discharge (50th & 90th). The blue solid lines represent the spread of ensemble i.e. 5th & 95th quantiles of the conditional discharge (50th quantile). Similarly, black solid lines represent the 5th & 95th quantiles of the conditional discharge (90th quantile). Right: Conditional discharge plot for 50th & 95th quantile of surge ((b) HBV and (d) SPHY). The blue & black band represents the area between the 50th & 90th quantile for 50th & 90th quantiles of conditional discharge. The vertical red dash line represents the lag zero.



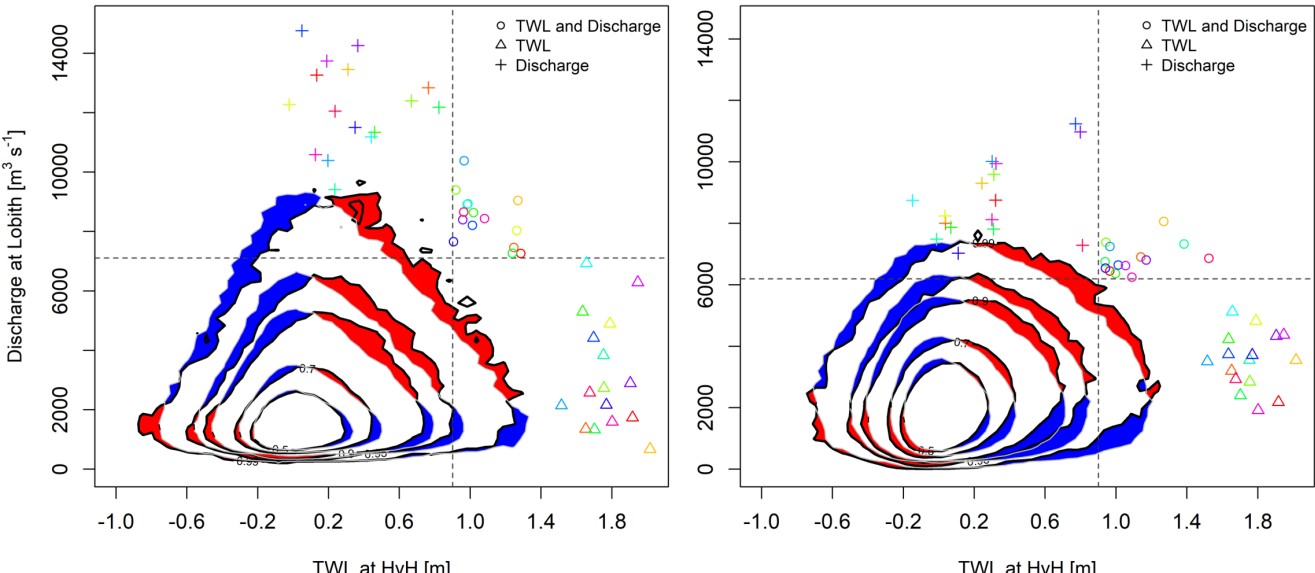

**Figure 7:** Joint probability density for total water level at HvH and discharge at Lobith (3-day lagged) for winter six months period (left-HBV and right-SPHY). The contours show different quantiles of the joint distribution (50th, 70th, 90th, 95th and 99th). Shading is used to contrast density estimates from direct model output and randomized (shuffled) data, where correlations have been artificially removed. Red/blue shading indicates regions where model data is more/less populated than the shuffled data. The thin dashed lines show the 99% quantiles of each variable in the respective dataset. The colored points correspond to the highest TWL (Δ), discharge (+) and compound events (o) for individual ensemble members.





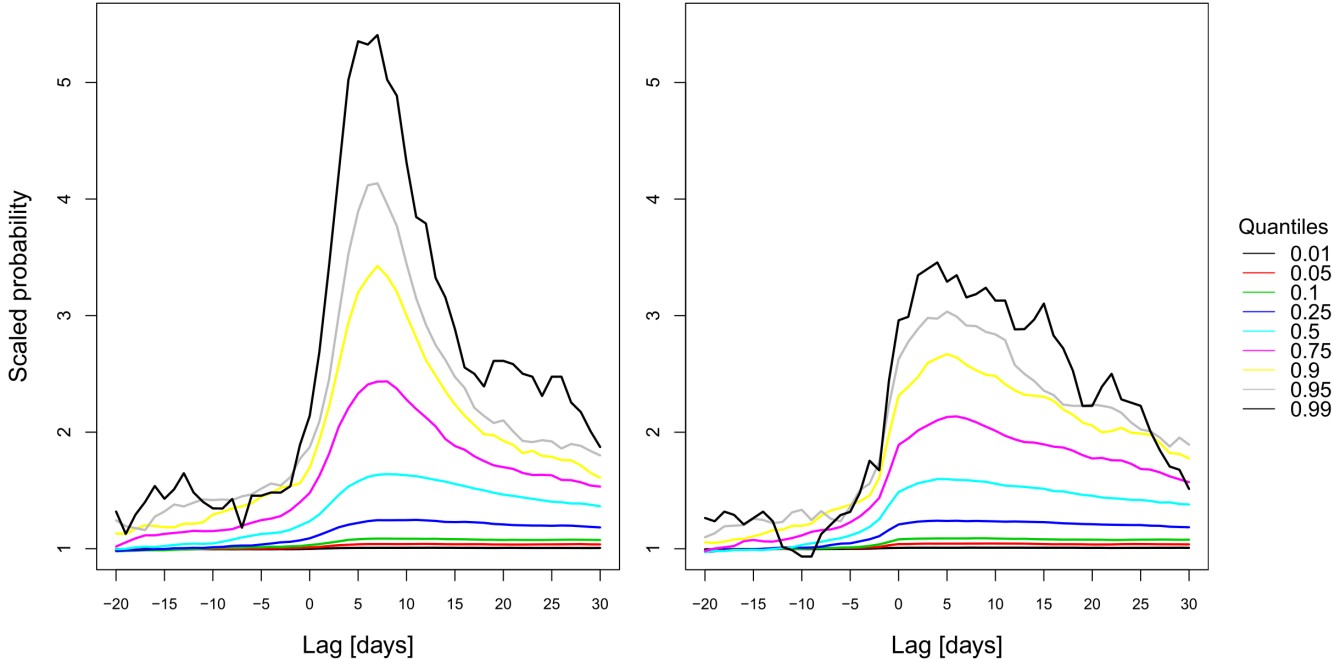

**Figure 8:** Probability for getting river discharge above a certain percentile (HBV- Left and SPHY- Right), given that total water levels also exceed the 97.5% percentile. For each discharge percentile, the probability is scaled by the random probability of the event.



**Table 1.** Performance index for HBV and SPHY model on a daily time scale. The low, med and high represent the statistics for Q<Q5th, Q5th < Q < Q95th and Q > Q95th quantile of the observed flow whereas, all, represents the overall flow series.

|  | **HBV** | | | | **SPHY** | | | |
|---|---|---|---|---|---|---|---|---|
| **Objective function** | low | Med | high | all | low | med | high | all |
| **R2** | 0.52 | 0.87 | 0.65 | 0.91 | 0.11 | 0.66 | 0.37 | 0.73 |
| **PBIAS (%)** | -18.3 | -10.6 | -7.3 | -10.3 | -10.9 | 4.8 | -20 | 1.3 |
| **RMSE(m3/s)** | 180 | 359 | 1045 | 415 | 352 | 534 | 1516 | 615 |
| **NSE** | -4.9 | 0.79 | 0.26 | 0.87 | -7.3 | 0.53 | -0.56 | 0.72 |
| **Volumetric Efficiency** | 0.82 | 0.87 | 0.85 | 0.87 | 0.69 | 0.82 | 0.78 | 0.81 |

