# Peer review of "Figure S1: Sub-basin discretization for HBV model (left, Huiskes 2016) and SPHY(right). HBV model used 134 sub basins to calibrate the model whereas, SPHY model was calibrated at 7 sub basins. The major sub-basin in the model are delineated at the same location."

_Hydrology and Earth System Sciences, 2018_

## Referee Comment (RC1) · Anonymous Referee #1 · 29 Jul 2018

This paper investigates statistical dependency between extreme river discharge and coastal water level in Rhine river basin. While the authors mainly followed a methodology established by van den Hurk et al. 2015 and Klerk et al. 2015, this study provides a unique contribution in that they used a large set of ensemble model simulation results, not just observations. I think the authors have conducted substantial amount of work and critically analyzed their results, the paper is well written for readers to easily follow, and the findings are scientifically new and interesting. Therefore, I recommend this paper to go through minor revisions before publication. Minor comments are listed below.

P 2, L 10: Underestimation of what?

P 3, L 3 and after: Please use n-dash (–) not hyphen (-) to indicate certain range of

values.

P 4, L 5: Add the full name before the abbreviation for TWL.

P 5, L 21: What is E-OBS?

P 6, L 3 and after: The3 in the unit m3/s should be superscript.

P 6, L4 and L 11: Both 'modeled' and 'modelled' are used throughout the paper, so use either of them.

P 6, L 20, 25–26 and 31–32: I agree with your rationale to use two hydrological models to assess model uncertainty as mentioned in P 4, L 8. However, as introduced here, SPHY is strongly biased in reproducing high discharges and HVB performs much better than SPHY. In the supplementary figure 4S, it is shown that SPHY's performance was better than HVB, but given that this paper's objective is to see the dependence between extreme values, it does not support the reason to use SPHY. I am not sure why the authors use such different models in terms of model types (i.e., SPHY is a conceptual model while HVB is a semi-distributed model) and the model physics (written in P 4, L 30–31) for comparison.

P 7, L 23–24: Why does SPHY have multiple maxima?

P 8, L 3–4: 'The broad shape of the distribution of both HBV and SPHY reflects the complex interaction of the climatic and hydrologic processes.' This sentence is too concise to understand the meaning. Why can you say that the broad distribution reflects the complexity in climatology-hydrology interactions? It would be helpful if you can add some more explanations.

P 8, L 8: Looking at Figure 4, the half of the data was located in the range -1–+1 in case of HVB, which does not seem so broad a distribution. P 9, L 3–4: Again, the SPHY results are strongly affected by the underestimation of river discharge. I am not sure whether the use of such a poorly biased model can provide meaningful indications.

P 9, L 12–13: The word 'tail' is duplicated in the sentence 'analyzing the tails of the tail of distributions'.

P 9, L 29: What does the width of the bands represent?

P 10, L 13–15: This analysis is interesting, but could you add some literature to support your reasoning about the hydrological characteristics of the target basin?

P 10, L 26: 'in which the physical. . .' maybe 'physics' not 'physical'?

P 12, L 18: 'still it's not. . .' The abbreviation should be avoided.

P 18, Figure 1 and after: Add the model names to each sub plot.

P 19, Figure 2: Maybe better to use 'and' instead of '&'.

P 19, Figure 3: What do the dotted lines in the right figure represent? Mean values?

P 22, Figure 6: In the scatter plots of the left figure, the blue dots represent events exceeding 99th quantile, but on the other hand, the black and blue lines in the left and right figures represent 95th and 50th quantiles, so the large/small relationship between gray and blue colors is inverse within the same figure. This is very confusing! In the right figures, what the triangular and the bands represent?

---

## Referee Comment (RC2) · Anonymous Referee #2 · 18 Sep 2018

Thank you for the opportunity to review this manuscript. This study deals with an important issue on compound flooding. It is very interesting to read. But there are a number of points that need to be clarified.

Detailed comments:

1. The authors presented a modeling framework involving a few different modeling platforms and an integration of them to examine the dependence between large riverine discharge and high sea level due to tide/surge. It will be good if the authors could illustrate the different modelling components, how they connect to each other, the flow of data/information in and out of them in a figure/chart. This will help the readers to underhand the entire modeling framework.

[Figure]

2. The authors mentioned that "For the Rhine catchment and the Dutch coastal area, existing studies suggest that no such relation is present at time lags shorter than six days." However, the outcomes of this current study contradict the previous findings. What are the mechanisms that have changed the outcomes of this current study?

3. As the authors noted that timing is important in dependence analysis. Both hydrological models "have errors in estimating the timings of flood waves" and "Both models have difficulties in reproducing flood timing". How these errors in timing actually contributed to the dependence estimated in this study compared to previous studies using observed data? Is it possible that the increased dependence found in this study is actually due to these errors in timing, rather than being genuine increased dependence?

4. Figure 6 and Figure 7 are important for this study. It seems the dependence between the total water level and river discharge is not very strong if there is any dependence at all. As shown in Figure 6, in the majority of the cases, the large values of the two processes occur independent of each other. It is not immediately clear to me how the authors arrived at the conclusions that "the probability for finding a co-occurrence of extreme river discharge at Lobith and storm surge conditions at Hoek van Holland are up to four times higher . . . ". This conclusion is only mentioned once in the abstract and cannot be found (and not explained) anywhere else in the paper.

5. By looking at Figure 6, the two processes seem to be asymptotically independent. Does this mean the driving forces of the two processes may change? For example, the extreme sea level may be driven by storm surge however, the very extreme sea levels maybe driven by astronomical tide, which is not correlated to river discharge? How will this impact the total dependence between the two processes?

6. How does the method used in this study compare to those ones commonly used in previous studies to assess the dependence between two variables, e.g. correlation-based (Wahl et al. 2015) or copula-based methods (Bevacqua et al. 2017), or the bivariate logistic threshold excess model (Wu et al. 2018)? Will there be similar/weaker/stronger dependence between the two processes?

7. Dependence between storm surge and other flood contributing processes have been studies previously (Svensson and Jones 2004; Wahl et al. 2015; Wu et al. 2018). However, this paper does not directly deal with storm surge; rather total water level was used. The title and heading of section 4 should be changed to reflect what was done in this study. In addition, Line 10, page 3: "We believe that the use of a large sample of data obtained from a fine resolution climate model ensemble provides a better insight into the statistical connections . . .". Similar recommendations were made/demonstrated in previous studies(Wahl et al. 2015; Wu et al. 2018)

Bevacqua, E., Maraun, D., Hobæk Haff, I., Widmann, M., and Vrac, M. (2017). "Multivariate statistical modelling of compound events via pair-copula constructions: analysis of floods in Ravenna (Italy)." Hydrol. Earth Syst. Sci., 21(6), 2701-2723. Svensson, C., and Jones, D. A. (2004). "Dependence between sea surge, river flow and precipitation in south and west Britain." Hydrol. Earth Syst. Sci., 8(5), 973-992. Wahl, T., Jain, S., Bender, J., Meyers, S. D., and Luther, M. E. (2015). "Increasing risk of compound flooding from storm surge and rainfall for major US cities." Nature Climate Change, 5, 1093. Wu, W., McInnes, K., O'Grady, J., Hoeke, R., Leonard, M., and Westra, S. (2018). "Mapping Dependence Between Extreme Rainfall and Storm Surge." Journal of Geophysical Research: Oceans, 123(4), 2461-2474.

Technical corrections

1. Line 27 page 2: country of US and Italy should swap as the first study by Bevacqua et al. (2017) is on Italy.

2. Line 29, page 7: "we chose it to use $\pm5$ days here" should be "we chose to use $\pm5$ days here".

3. Line 12 page 9. Pleas reword the last sentence.
* * *
103, 2018.

---

## Author Comment (AC1) · 16 Oct 2018

**Received and published: 29 July 2018**

This paper investigates statistical dependency between extreme river discharge and coastal water level in Rhine river basin. While the authors mainly followed a methodology established by van den Hurk et al. 2015 and Klerk et al. 2015, this study provides a unique contribution in that they used a large set of ensemble model simulation results, not just observations. I think the authors have conducted substantial amount of work and critically analyzed their results, the paper is well written for readers to easily follow, and the findings are scientifically new and interesting. Therefore, I recommend this paper to go through minor revisions before publication. Minor comments are listed below.

We are grateful to the reviewer for the thorough review of our manuscript and useful suggestions, which improved the quality of the manuscript significantly. We provide a point-by-point clarification and response to the reviewer's comments below. For clarity, the reviewer's comments are given in red color, the responses are given in plain black text and the modifications in the manuscript are in blue italics. The manuscript will be modified accordingly.

**P 2, L 10: Underestimation of what?**

This is a typographic error, and the sentence should be completed as "Ignoring the dependencies may lead to severe over or underestimation of the flood risk". It will be corrected in the revised manuscript.

**P2, L10 "Ignoring the dependencies may lead to severe over or under estimation of the flood risk".**

**P 3, L 3 and after: Please use n-dash (-) not hyphen (-) to indicate certain range of**

This will be corrected throughout the manuscript.

**P 4, L 5: Add the full name before the abbreviation for TWL.**

This will be added.

**P 5, L 21: What is E-OBS?**

E-OBS are the observed gridded daily precipitation data sets available for European region. Since the data user agreement requires it to be abbreviated as "E-OBS", we use the same acronym and have described this dataset previously in P 3, L30 in original manuscript.

**P 6, L 3 and after: The3 in the unit m3/s should be superscript.**

We will correct this.

**P 6, L4 and L 11: Both 'modeled' and 'modelled' are used throughout the paper, so use either of them.**

We will use "modeled" consistently throughout the revised manuscript.

P 6, L 20, 25–26 and 31–32: I agree with your rationale to use two hydrological models to assess model uncertainty as mentioned in P 4, L 8. However, as introduced here, SPHY is strongly biased in reproducing high discharges and HVB performs much better than SPHY. In the supplementary figure 4S, it is shown that SPHY's performance was better than HVB, but given that this paper's objective is to see the dependence between extreme values, it does not support the reason to use SPHY. I am not sure why the authors use such different models in terms of model types (i.e., SPHY is a

**conceptual model while HVB is a semi-distributed model) and the model physics (written in P 4, L30–31) for comparison.**

We acknowledge the reviewer's concern regarding the underestimation of the extremes in the SPHY model. The rationale to use two hydrological models is to include the uncertainty from using different hydrological models. The large bias in SPHY was a clear motivation to include an evaluation of the HBV model as well. However, later recalibration of the SPHY model has led do a clearly improved performance of this model. All SPHY results are replaced and modified conclusions have been included accordingly. The modification to the main results and conclusions are described in more detail in the reply to comments below.

Other aspects and major conclusions remain unchanged. We still point at the need to analyse flood wave length and timing in detail before analyzing the joint flooding risk. We show that flood wave timing is of prime importance while assessing the coastal flood risk. Neither of the models used in this study have perfect performance. The recalibrated version of SPHY outperforms HBV in the representation of the mean annual cycle and daily biases were lower in SPHY. Although HBV performs fairly well in representing the extremes, the flood wave timing was not perfect. The results of two hydrological models with different performance allows evaluation of the impact of these bias on the correlation characteristics, and gives an indication of the contribution of model bias to uncertainty of this joint correlation. Since the results presented in this study are based on quantile thresholds relative to the respective dataset, the biases in the model results do impact the findings concerning the statistical relation between water level and river discharge.

**P 7, L 23–24: Why does SPHY have multiple maxima?**

The phrasing "multiple maxima" is bit misleading. We intended to say that the maxima is broader in SPHY and is not always a well-defined peak as in the observations. This is the motivation to use the onset of flood wave rather than maxima. However, later recalibration of the SPHY model for the revised manuscript has led do a clearly improved performance of this model. With the improved simulation of extreme flows, more well-defined flood peaks are achieved in the SPHY model simulation. Rather than comparing the onset of discharge peaks, we directly compare the flood wave peak timing. The modification to the main results and conclusions are described in more detail in the reply to comments below.

P 8, L 3–4: 'The broad shape of the distribution of both HBV and SPHY reflects the complex interaction of the climatic and hydrologic processes.' This sentence is too concise to understand the meaning. Why can you say that the broad distribution reflects the complexity in climatology-hydrology interactions? It would be helpful if you can add some more explanations.

We added explanation for this statement in the revised manuscript: This wide distribution is a result of multiple drivers of flood rather than a single flood generation mechanism. The climatic mechanism includes persistent synoptic weather conditions favoring a very extreme event or episodes of moderate precipitation events resulting in a multiple day extreme event or extreme positive temperature anomaly causing a quick melt of snow in the catchment (Gaál et al., 2012; Nied et al., 2014; Prudhomme and Genevier, 2011). The hydrological processes such as antecedent soil moisture conditions, snow and ice storage in the catchment, rain on snow mechanisms and antecedent ground water level play an important role in defining the magnitude and length of the flood wave (Merz and Blöschl, 2003, 2008). Further, the superimposition of flood waves from different tributaries of the river also contributes towards the increased length and magnitude of the flood wave. Moreover, coincidence of any of the extremes from the climatic and hydrological processes results in amplification of the flooding magnitude and extent. P 8, L 8: Looking at Figure 4, the half of the data was located in the range -1-+1 in case of HVB, which does not seem so broad a distribution. P 9, L 3–4: Again, the SPHY results are strongly affected by the underestimation of river discharge. I am not sure whether the use of such a poorly biased model can provide meaningful indications.

We acknowledge the reviewer's concern regarding the large biases in reproducing the absolute extremes in the SPHY model. However, after recalibration the results have changed strongly as indicated above.

Since, the reviewers have some concern over the SPHY model results, we decided to couple an advanced kinematic routing scheme to the model which improves the representation of the flood wave characteristics, and allows a better quantification of the role of model uncertainty. We use the PCR-GLOWB2 kinematic wave routing scheme (Sutanudjaja et al., 2018). The higher quantiles flow has significantly improved as compared the SPHY with simple routing scheme (Figure R1 and Table 1.)

Figure R1. Observed versus SPHY modeled daily discharge at Lobith for the period between 1951 to 2000 for (left) the original simple routing scheme, and (right) with a kinematic routing scheme from PCR-GLOWB-2. Colors indicate three ranges based on observed percentiles: "Low" (<5%, red), "Medium" (5-95%, green) and "High" (>95%). The solid red line represents the 1:1 slope.

**Table 1.** Performance index for HBV and SPHY model on a daily time scale. The low, med and high represent the statistics for Q<Q5th, Q5th < Q < Q95th and Q > Q95th quantile of the observed flow whereas, all, represents the overall flow series.

HBV

| Objective function    | low   | Med   | high | all   | low   | med  | high | all  |
|-----------------------|-------|-------|------|-------|-------|------|------|------|
| R2                    | 0.52  | 0.87  | 0.65 | 0.91  | 0.19  | 0.65 | 0.34 | 0.77 |
| PBIAS (%)             | -18.3 | -10.6 | -7.3 | -10.3 | -20   | -0.1 | 6.7  | 0.3  |
| RMSE(m3/s)            | 180   | 359   | 1045 | 415   | 300   | 605  | 1732 | 695  |
| NSE                   | -4.9  | 0.79  | 0.26 | 0.87  | -5.26 | 0.59 | 0.2  | 0.69 |
| Volumetric Efficiency | 0.82  | 0.87  | 0.85 | 0.87  | 0.72  | 0.81 | 0.79 | 0.81 |

Figure R2: Normal quantile plot for HBV (blue), SPHY (red) and Observation (black). On the horizontal axis, the distributions are centered and scaled (divided by the standard deviation). The light blue & red lines represent 16 ensemble members for HBV & SPHY.

With the improved simulation of extreme flows, more well-defined flood peaks are observed in the model simulation. Rather than comparing the onset of discharge peaks, we directly compare the flood wave peak timing. We found that with the new routing scheme the flood wave travel time has significantly improved in SPHY, even outperforming the HBV model (Figure R3).

---

## Author Comment (AC2) · 16 Oct 2018

Thank you for the opportunity to review this manuscript. This study deals with an important issue on compound flooding. It is very interesting to read. But there are a number of points that need to be clarified.

We are grateful to the reviewer for reviewing our manuscript carefully and for his/her useful suggestions, which improved the quality of the manuscript significantly. We provide a point-by-point clarification and response to the reviewer's comments below. For clarity, the reviewer's comments are given in red color, the responses are given in plain black text and the modification/addition in the manuscript are in blue italics. The revised manuscript will be modified accordingly.

Detailed comments:

The authors presented a modeling framework involving a few different modeling platforms and an integration of them to examine the dependence between large riverine discharge and high sea level due to tide/surge. It will be good if the authors could illustrate the different modelling components, how they connect to each other, the flow of data/information in and out of them in a figure/chart. This will help the readers to underhand the entire modeling framework.

This is a useful suggestion. A figure with schematic representation of the modeling framework will be added to the supplementary document to illustrate the inputs, models and outputs in this study.

Modelling frame work

[Figure]

Figure R1. The framework used in this study. The orange color, green and grey color represents the input, output and the model used in the study respectively. The P, T, SLP, U and V represents the precipitation, temperature, sea level pressure, zonal and meridional wind components respectively. The blue color represents the intermediate steps followed like bias correction of precipitation and lapse rate correction of temperature obtained from downscaling EC-Earth.

2. The authors mentioned that "For the Rhine catchment and the Dutch coastal area, existing studies suggest that no such relation is present at time lags shorter than six days." However, the outcomes of this current study contradict the previous findings. What are the mechanisms that have changed the outcomes of this current study?

A straightforward rationale exists to link high storm surges to large discharge volumes arriving at the coast 6 days later: a strong storm whose winds set up a storm surge will need time to reach the Rhine headwaters where heavy rainfall will find its way to the river mouth after a few days of travel. However, in the real world this simple rationale is blurred by natural variability where multiple storms and anomalous travel times may lead to very different correlation lag times. Our ensemble high-resolution model set-up allows diagnosing this correlation range much better than studies that rely on single observational records (Figure R2).

[Figure]

Figure R2: Mean temporal evolution of the 75th (black) 90th (red) and 95th (blue) quantile of observed discharge at Lobith for total water level events exceeding the 90th percentile at HvH for observations. The dash line and solid lines represent the unconditional discharge and conditional discharge on total water level respectively.

3. As the authors noted that timing is important in dependence analysis. Both hydrological models "have errors in estimating the timings of flood waves" and "Both models have difficulties in reproducing flood timing". How these errors in timing actually contributed to the dependence estimated in this study compared to previous studies using observed data? Is it possible that the increased dependence found in this study is actually due to these errors in timing, rather than being genuine increased dependence?

We acknowledge the reviewers concern regarding the impact of model biases in the results of this study. We agree that a major part of the uncertainty is coming from model errors. However, restricting ourselves to only use observations would restrict us to a smaller pool of samples. As discussed above, the correlation may emerge at other lags than 6 days, due to natural variability in

the system. We also acknowledge that the broad simulated flood waves in SPHY might have contributed towards the increased dependency. For instance, in SPHY at higher quantiles the confidence band are overlapping with unconditional line (Figure 5 P 21 in original manuscript), statistically no conclusions can be drawn. Flood wave timing and biases in the extreme events might have smeared out the signal in the surge discharge composite. Nevertheless, a clear correlation can be seen at lower quantiles.

Since, the reviewers have some concern over the SPHY model results, we decided to couple an advanced kinematic routing scheme to the model which improves the representation of the flood wave characteristics, and allows a better quantification of the role of model uncertainty. We use the PCR-GLOWB2 kinematic wave routing scheme (Sutanudjaja et al., 2018). The higher quantiles flow has significantly improved as compared the SPHY with simple routing scheme (Figure R3 and Table 1.)

[Figure]

Figure R3. Observed versus modeled daily discharge at Lobith for the period between 1951 to 2000 for SPHY model left for the previous simple routing scheme as described in the manuscript and right with kinematic routing scheme from PCR-GLOWB-2. Colors indicate three ranges based on observed percentiles: "Low" (<5%, red), "Medium" (5-95%, green) and "High" (>95%). The solid red line represents the 1:1 slope.

**Table 1.** Performance index for HBV and SPHY model on a daily time scale. The low, med and high represent the statistics for Q<Q5th, Q5th < Q < Q95th and Q > Q95th quantile of the observed flow whereas, all, represents the overall flow series.

| | HBV | | | | SPHY (Kinematic routing) | | | |
|---|---|---|---|---|---|---|---|---|
| **Objective function** | low | Med | high | all | low | med | high | all |
| **R2** | 0.52 | 0.87 | 0.65 | 0.91 | 0.19 | 0.65 | 0.34 | 0.77 |
| **PBIAS (%)** | -18.3 | -10.6 | -7.3 | -10.3 | -20 | -0.1 | 6.7 | 0.3 |
| **RMSE(m3/s)** | 180 | 359 | 1045 | 415 | 300 | 605 | 1732 | 695 |
| **NSE** | -4.9 | 0.79 | 0.26 | 0.87 | -5.26 | 0.59 | 0.2 | 0.69 |
| **Volumetric Efficiency** | 0.82 | 0.87 | 0.85 | 0.87 | 0.72 | 0.81 | 0.79 | 0.81 |

[Figure]

Figure R4: Normal quantile plot for HBV (blue), SPHY (red) and Observation (black). On the horizontal axis, the distributions are centered and scaled (divided by the standard deviation). The light blue & red lines represent 16 ensemble members for HBV & SPHY.

With the improvement of flow in higher quantiles, a more well-defined flood peaks are observed in the model simulation. Rather than comparing the onset of discharge peaks, we directly compare the flood wave peak timing. We found that the with the new routing scheme the flood wave travel time has significantly improved, even better than the HBV model (Figure R5).

[Figure]

Figure R5. Comparison of the timing distribution of the discharge wave peak in SPHY and as compared with the observations for 50 years.

With the model routing improvement, the surge and discharge composite plots generated using the climate model ensemble are also improved, as shown in Figure R6. This suggest the underestimation of the extreme flows in SPHY are mainly due to routing scheme used and not due to poor calibration of the physical processes. With improved timings of the flood wave, the model uncertainty can be reduced further. A clear dependence at higher quantile for the range of lags can be seen in SPHY model with kinematic routing which was not evident in the SPHY model with simple routing. Improvement in the model timing does not change our previous claim that probability for finding a co-occurrence of extreme river discharge at Lobith and storm surge conditions at Hoek van Holland are up to four times higher (Figure R7). Though there are some minor differences in the figures, the main conclusion remains unchanged.

*Based on this, we change section 3.1.1 Basic metrics and distribution, 3.1.3 Timing of the peak, 5. Discussion, table 1, and all the figures 1-8 accordingly in the main manuscript.*

[Figure]

[Figure]

Figure R6. Mean temporal evolution of the 90[th] (red), 95[th] (green) and 99[th] (blue) quantile of discharge at Lobith for total water level events exceeding the 90th percentile at HvH in WAQUA as modeled by (a) HBV, (b)SPHY with simple routing and (c) SPHY with kinematic routing. The lag in discharge at Lobith is relative to the peak in total sea water level at HvH. Negative & positive lag days indicate that the discharge peak occurs before & after the day of the high sea water event. The dashed lines are the unconditional discharge quantiles, i.e. discharge quantiles independent of water level; solid lines are the ensemble mean of the conditional quantiles. The shaded area represents 16 different lines for each ensemble and we only took the 5[th] and 95[th] percentile of those 16 lines to show spread of 16 ensemble member.

[Figure]

[Figure]

Figure R7. Probability for getting river discharge above a certain percentile ((a) HBV, (b)SPHY with simple routing and (c) SPHY with kinematic routing), given that total water levels also exceed the 97.5% percentile. For each discharge percentile, the probability is scaled by the random probability of the event.

4. Figure 6 and Figure 7 are important for this study. It seems the dependence between the total water level and river discharge is not very strong if there is any dependence at all. As shown in Figure 6, in the majority of the cases, the large values of the two processes occur independent of each other. It is not immediately clear to me how the authors arrived at the conclusions that "the probability for finding a co-occurrence of extreme river discharge at Lobith and storm surge conditions at Hoek van Holland are up to four times higher ... ". This conclusion is only mentioned once in the abstract and cannot be found (and not explained) anywhere else in the paper.

We agree with the reviewer, that we do not see strong dependence between the two variables. We also agree that for large values, the extremes in the two variables are virtually independent of each other. However, in our study we are mainly interested in the understanding of how the conditional discharges on total water level (TWL) vary with respect to the unconditional discharges (Figure 6 P22 in original manuscript). The background scatter plots serve as an illustration of the joint distribution. The conclusion "the probability for finding a co-occurrence of extreme river discharge at Lobith and storm surge conditions at Hoek van Holland are up to four times higher ..."  is made based on the results obtained from section 4.3 P11 and Figure 8 P24 in the original manuscript. In this section, we assess the compound event probabilities scaled with the unconditional probabilities. Based on the results, we found that at very high quantiles the normalized probability strongly increases. Depending upon the model, we found that it is 2-5 times more likely to get a high discharge once TWL is high.

Based on this, we include the following in section 4.1 P9 L20

*"The background scatter point only serves as an illustration of the joint distribution of the two variables. Presence of red scatter points on the top right corner implies the dependence between two variables. However, these two variables do not show any strong dependence, especially for very large value."*

5. By looking at Figure 6, the two processes seem to be asymptotically independent. Does this mean the driving forces of the two processes may change? For example, the extreme sea level may be driven by storm surge however, the very extreme sea levels maybe driven by astronomical tide, which is not correlated to river discharge? How will this impact the total dependence between the two processes?

We acknowledge the reviewer's concern that extreme sea levels may be driven by astronomical tide rather than the storm surge. To understand this, we conducted a similar analysis as in Figure 6(a) using only the storm surge levels (Figure R7). We do not find any significant differences between the two results except the position of the points. The statistical properties relevant for this study are similar. Since we use the daily mean TWL values, the tide signals are averaged out. If we would have taken the daily maximum values for tide and surge then the largest peak would occur when storm surge and tide coincide. There is no such extreme case which is dominated by the astronomical tide only for Dutch coastal area.

[Figure]

Figure R7. Scatter plot of coastal water levels and discharge for a lag of three days ((a) HBV and (b) SPHY with simple routing and (c) SPHY with kinematic routing) and 16 ensemble members. Events exceeding the 99th quantile of either of the variables are marked in blue. Events exceeding the 99th quantile of both variables are marked in red. The triangles (blue/black) represent the ensemble mean of the conditional discharge (50th & 90th). The blue solid lines represent the spread of ensemble i.e. 5th & 95th quantiles of the conditional discharge (50th quantile). Similarly, black solid lines represent the 5th & 95th quantiles of the conditional discharge (90th quantile).

Based on this, we include the following in section 4 P8 L19

*"We take daily mean TWL instead of the max TWL to eliminate any possibilities of extreme sea levels that may be driven by astronomical tide. Since by using the daily mean TWL, the effect of astronomical tide is averaged out. Further, there is no such extreme case which is dominated by the astronomical tide only for Dutch coastal area. Thus, we define a high/extreme water event as a day where the daily mean TWL at HvH exceeds the 90th percentile of its distribution."*

6. How does the method used in this study compare to those ones commonly used in previous studies to assess the dependence between two variables, e.g. correlation based (Wahl et al. 2015) or copula-based methods (Bevacqua et al. 2017), or the bivariate logistic threshold excess model (Wu et al. 2018)? Will there be similar/weaker/stronger dependence between the two processes?

We acknowledge the reviewers concern regarding the comparison of the method used in this study as compared to the studies mentioned by the reviewer (Table R1). The study from Wahl et al., 2015, focuses on the precipitation and storm surge along the US coasts. Their study uses precipitation as proxy for riverine floods. Further, the dependence structure is modeled using a family of copula models. Similarly, the study from Wu et al., 2018 used the observed rainfall and storm surge data to calculate the dependencies along the Australian coast. They used the bivariate logistic threshold excess model to calculate the dependence between the variables. Further, they used the hydrodynamic model to simulate the dependencies between the variables in a data sparse region with a reasonable level of confidence.

Both Wahl et al., 2015 and Wu et al., 2018 used precipitation (rainfall) and storm surge to investigate the coastal flood risk. However, discharge (water level) is a more appropriate variable than precipitation. This motivates this study to use hydrological model. Further, Bevacqua et al., 2017 investigated the uncertainties in flood risk with the use of pair copula constructions (PCCs) for Ravenna (Italy) using the water level as one of the impact variables. Their study focuses on PCCs to make a distinction between the heterogeneous dependence structure among different variables which is difficult to model using the multivariate parametric gaussian copula models. Since the multivariate gaussian copula assume marginal distribution for all the variables, the same dependency structure in all the generated pairs is not reasonable. To overcome this limitation, they use the PCCs. However, such copula construction lacks the memory components associated with the hydro-meteorological systems. The hydrological processes such as antecedent soil moisture conditions, snow and ice storage in the catchment, rain on snow mechanisms and antecedent ground water level which play an important role in defining the magnitude and length of the flood wave (Merz and Blöschl, 2003, 2008) are lacked in copula construction. Though, copula model are very useful and powerful tools in simulating the dependencies between the variables in multivariate environment, the statistical models rely on many assumptions to simulate the joint behavior or distribution. The joint distribution of the variables is difficult to know beforehand and rather than using (statistical) copula's that rely on many assumptions, we apply a physical modelling framework that generates the physical correlations directly.

Our study directly assesses the compound nature of water level and discharge, rather than indirect proxies. With the use of models and gridded climate data, we ensure that the heterogenous variation of meteo-hydrological processes and the memory components are well captured here which are lacking in the above point scale studies. We rather use a complete physical approach to show the joint occurrence of high discharge and water level are not just by a chance.

We acknowledge the reviewers concern regarding the strength of dependence with the use of different methods. Klerk et al., 2015 used the bivariate logistic threshold excess approach and showed no asymptotic behavior towards more extreme quantiles in two variables in the Rhine. However, the study is based on a relatively short period of data and a coarse climate model simulation. We did not assess the dependence structure with the copula based models yet as it is not in the scope of this study. The motivation of this study was to build upon previous studies from van den Hurk et al., 2015; Kew et al., 2013; Klerk et al., 2015 and physically understand the mechanism behind the increased dependence between the real impact variables. Further investigation using the statistical approaches mentioned by the reviewer would be interesting for future work.

Table R1. Table showing the data and the approaches used in the different studies mentioned by the reviewer.

|  | Wahl et al | Wu et al | Bevacqua et al | This study |
|---|---|---|---|---|
| **Variables** | Precipitation and Storm surge (without tidal and mean sea level influence) | Rainfall and Storm surge (nontidal residuals) | Water levels, sea levels | Total water level and River discharge |
| **Data** | Observations and modeled pair using copula | Observations and surges from hydrodynamic model | Observations and PCCs simulation | Dynamically downscaled EC-Earth model |

| | | | | data in ensemble mode. |
|---|---|---|---|---|
| **Methods** | Dependency simulation using families of copula models | Dependency between the rainfall and storm surge is simulated using the hydrodynamic model | Dependency is simulated using the PCCs. | Physical modelling of the surge and river component though hydrodynamic, hydrological and hydraulic model. |
| **Measure** | Kendall's rank correlation coefficient | Alpha ($\alpha$) parameter from the bivariate logistic threshold excess method is used to assess the dependency. | Uncertainty in estimation of impact variable (water level affected from river and sea water) together with meteorological predictors.s | Dependence is measured based on the joint distribution from large ensemble of correlated modeled pairs. The scaled probabilities of joint events are calculated as a measure of strength. |
| **Region** | US coast | Australian coast | Ravenna, Italy | Rhine (at Lobith) |

Based on this, we include the following in main manuscript after P13 L1.

*Though, copula model are very useful and powerful tools in simulating the dependencies between the variables in multivariate environment, the statistical models rely on many assumptions to simulate the joint behavior or distribution. The joint distribution of the variables is difficult to know beforehand and rather than using (statistical) copula's that rely on many assumptions, a physical modelling framework that generates the physical correlations directly is needed. To this end we adopt a methodology that directly assesses the compound nature of sea water level and discharge, rather than indirect proxies. With the use of sets of hydrological models and fine resolution gridded climate data in ensemble mode, we ensure that the heterogenous variation of meteo-hydrological processes and the memory components of the system are well captured here which are lacking in the studies before in Rhine. We rather used a complete physical approach to investigate the joint occurrence of high discharge and water level are not just by a chance.*

7. Dependence between storm surge and other flood contributing processes have been studies previously (Svensson and Jones 2004; Wahl et al. 2015; Wu et al.2018). However, this paper does not directly deal with storm surge; rather total water level was used. The title and heading of section 4 should be changed to reflect what was done in this study. In addition, Line 10, page 3: "We believe that the use of a large sample of data obtained from a fine resolution climate model ensemble provides a better insight into the statistical connections ...". Similar recommendations were made/demonstrated in previous studies (Wahl et al. 2015; Wu et al. 2018)

We refer to our response to comment no. 5 for the motivation to use "Storm surge" in the title instead of the Total water level

The sentence L10, P3 will be rephrased as suggested by the reviewer:

*"The use of large sample of data obtained from a fine resolution climate model ensemble provides a better insight in to the statistical connection between the two variables (Wahl et al., 2015; Wu et al., 2018) than possible in previous assessments in Rhine which used either samples from observations limited to the past 30-40 years data only or used limited year coarse resolution climate model simulations."*

Bevacqua, E., Maraun, D., Hobæk Haff, I., Widmann, M., and Vrac, M. (2017). "Multivariate statistical modelling of compound events via pair-copula constructions: analysis of floods in Ravenna (Italy)." Hydrol. Earth Syst. Sci., 21(6), 2701-2723.

Svensson, C.,and Jones, D. A. (2004). "Dependence between sea surge, river flow and precipitation in south and west Britain." Hydrol. Earth Syst. Sci., 8(5), 973-992.

Wahl, T., Jain,S., Bender, J., Meyers, S. D., and Luther, M. E. (2015). "Increasing risk of compound flooding from storm surge and rainfall for major US cities." Nature Climate Change,5, 1093.

Wu, W., McInnes, K., O'Grady, J., Hoeke, R., Leonard, M., and Westra, S.(2018). "Mapping Dependence Between Extreme Rainfall and Storm Surge." Journal of Geophysical Research: Oceans, 123(4), 2461-2474.

Technical corrections

1. Line 27 page 2: country of US and Italy should swap as the first study by Bevacqua et al. (2017) is on Italy.

This will be corrected in the revised manuscript.

2. Line 29, page 7: "we chose it to use ±5 days here" should be "we chose to use ±5 days here".

We agree with the reviewer. It will be corrected in the manuscript.

3. Line 12 page 9. Pleas reword the last sentence.

We agree with the reviewer. The sentence will be rephrased to *"The 800 years of daily data provides a solid confidence in analysing the tails of the distributions."*

References

Bevacqua, E., Maraun, D., Hobæk Haff, I., Widmann, M. and Vrac, M.: Multivariate statistical modelling of compound events via pair-copula constructions: Analysis of floods in Ravenna (Italy), Hydrol. Earth Syst. Sci., doi:10.5194/hess-21-2701-2017, 2017.

van den Hurk, B., van Meijgaard, E., de Valk, P., van Heeringen, K.-J. and Gooijer, J.: Analysis of a compounding surge and precipitation event in the Netherlands, Environ. Res. Lett., 10(3), 35001,

doi:10.1088/1748-9326/10/3/035001, 2015.

Kew, S. F., Selten, F. M., Lenderink, G. and Hazeleger, W.: The simultaneous occurrence of surge and discharge extremes for the Rhine delta, Nat. Hazards Earth Syst. Sci., 13(8), 2017–2029, doi:10.5194/nhess-13-2017-2013, 2013.

Klerk, W. J., Winsemius, H. C., van Verseveld, W. J., Bakker, A. M. R. and Diermanse, F. L. M.: The co-incidence of storm surges and extreme discharges within the Rhine–Meuse Delta, Environ. Res. Lett., 10(3), 35005, doi:10.1088/1748-9326/10/3/035005, 2015.

Merz, R. and Blöschl, G.: A process typology of regional floods, Water Resour. Res., 39(12), 1–20, doi:10.1029/2002WR001952, 2003.

Merz, R. and Blöschl, G.: Flood frequency hydrology: 1. Temporal, spatial, and causal expansion of information, Water Resour. Res., 44(8), 1–17, doi:10.1029/2007WR006744, 2008.

Sutanudjaja, E. H., Van Beek, R., Wanders, N., Wada, Y., Bosmans, J. H. C., Drost, N., Van Der Ent, R. J., De Graaf, I. E. M., Hoch, J. M., De Jong, K., Karssenberg, D., López López, P., Peßenteiner, S., Schmitz, O., Straatsma, M. W., Vannametee, E., Wisser, D. and Bierkens, M. F. P.: PCR-GLOBWB 2: A 5 arcmin global hydrological and water resources model, Geosci. Model Dev., 11(6), 2429–2453, doi:10.5194/gmd-11-2429-2018, 2018.

Wahl, T., Jain, S., Bender, J., Meyers, S. D. and Luther, M. E.: Increasing risk of compound flooding from storm surge and rainfall for major US cities, Nat. Clim. Chang., 5(July), doi:10.1038/nclimate2736, 2015.

Wu, W., McInnes, K., O'Grady, J., Hoeke, R., Leonard, M. and Westra, S.: Mapping Dependence Between Extreme Rainfall and Storm Surge, J. Geophys. Res. Ocean., 123(4), 2461–2474, doi:10.1002/2017JC013472, 2018.